# Circulating Trimethylamine-N-Oxide Is Elevated in Liver Transplant Recipients

**DOI:** 10.3390/ijms25116031

**Published:** 2024-05-30

**Authors:** Maria Camila Trillos-Almanza, Mateo Chvatal-Medina, Margery A. Connelly, Han Moshage, Stephan J. L. Bakker, Vincent E. de Meijer, Hans Blokzijl, Robin P. F. Dullaart

**Affiliations:** 1Department of Gastroenterology and Hepatology, University Medical Center Groningen, University of Groningen, P.O. Box 30.001, 9700 RB Groningen, The Netherlands; mateo.medina@udea.edu.co (M.C.-M.); a.j.moshage@umcg.nl (H.M.); h.blokzijl@umcg.nl (H.B.); 2Labcorp, 100 Perimeter Park, Morrisville, NC 27560, USA; connem5@labcorp.com; 3Groningen Institute for Organ Transplantation, University Medical Center Groningen, University of Groningen, 9700 AD Groningen, The Netherlands; datarequest.transplantlines@umcg.nl; 4Department of Internal Medicine, Division of Nephrology, University Medical Center Groningen, University of Groningen, P.O. Box 30.001, 9700 RB Groningen, The Netherlands; s.j.l.bakker@umcg.nl; 5Department of Surgery, Division of Hepato-Pancreato-Biliary Surgery and Liver Transplantation, University Medical Center Groningen, University of Groningen, P.O. Box 30.001, 9700 RB Groningen, The Netherlands; v.e.de.meijer@umcg.nl; 6Department of Internal Medicine, Division of Endocrinology, University Medical Center Groningen, University of Groningen, P.O. Box 30.001, 9700 RB Groningen, The Netherlands; dull.fam@12move.nl

**Keywords:** liver transplantation, microbiota, NMR spectroscopy, Prevention of REnal and Vascular ENd-stage Disease (PREVEND) cohort study, TransplantLines cohort and biobank study, trimethylamine-N-oxide

## Abstract

Liver transplant recipients (LTRs) have lower long-term survival rates compared with the general population. This underscores the necessity for developing biomarkers to assess post-transplantation mortality. Here we compared plasma trimethylamine-N-oxide (TMAO) levels with those in the general population, investigated its determinants, and interrogated its association with all-cause mortality in stable LTRs. Plasma TMAO was measured in 367 stable LTRs from the TransplantLines cohort (NCT03272841) and in 4837 participants from the population-based PREVEND cohort. TMAO levels were 35% higher in LTRs compared with PREVEND participants (4.3 vs. 3.2 µmol/L, *p* < 0.001). Specifically, TMAO was elevated in LTRs with metabolic dysfunction-associated steatotic liver disease, alcohol-associated liver disease, and polycystic liver disease as underlying etiology (*p* < 0.001 for each). Among LTRs, TMAO levels were independently associated with eGFR (std. β = −0.43, *p* < 0.001) and iron supplementation (std. β = 0.13, *p* = 0.008), and were associated with mortality (29 deaths during 8.6 years follow-up; log-rank test *p* = 0.017; hazard ratio of highest vs. lowest tertile 4.14, *p* = 0.007). In conclusion, plasma TMAO is likely elevated in stable LTRs, with impaired eGFR and iron supplementation as potential contributory factors. Our preliminary findings raise the possibility that plasma TMAO could contribute to increased mortality risk in such patients, but this need to be validated through a series of rigorous and methodical studies.

## 1. Introduction

Liver transplantation (LT) is a well-established therapy for patients with acute and chronic end-stage liver diseases [1]. Recent reports indicate that the 5-year survival rate after LT exceeds 70%, followed by a decrease to 50–60% survival at 10 years [2,3,4,5]. This implies that long-term survival in LT recipients (LTRs) still remains compromised when compared with the general population, which is mainly due to cardiovascular and metabolic burdens, and residual malignant and infectious diseases [6,7,8]. Current focus is on inflammatory and tumoral biomarkers [9,10,11], but there is an unmet need of novel cardiovascular and metabolic biomarkers more precisely, with the aim to improve morbidity and decrease mortality in this population.

In recent years, there has been increasing focus on the gut–liver axis, particularly on the impact of the gut microbiome on liver diseases. The liver is inherently exposed to microbiota-derived metabolites as a first-pass organ due to the portal circulation, and bacterial translocation may therefore play a role in chronic liver diseases [12,13]. Within this context, a potential biomarker is the gut microbiome-derived metabolite, trimethylamine N-oxide (TMAO, N,N-dimethylmethanamine N-oxide). TMAO has been linked to several cardiovascular and metabolic disorders, including hypertension, atherosclerosis, coronary heart disease, chronic heart failure, chronic kidney disease, and diabetes [14,15,16,17,18,19,20,21]. In the general population, as well as in individuals with impaired renal function, diabetes, and MASLD, circulating TMAO is associated with all-cause and cardiovascular mortality [14,15,16,17,18,19,20,22,23,24,25].

TMAO is derived from trimethylamine (TMA, N,N-dimethylmethanamine), originating from the metabolism of choline, phosphatidylcholine, L-carnitine, and betaine in the gut [21]. TMA can be acquired from animal-based foods, including red meat, fish, and eggs. After it enters the portal circulation, TMA is oxidized into TMAO in the liver by flavin monooxygenase 3 (FMO3) [26,27]. Consequently, the circulating TMAO concentration is directly related to the availability of precursors in the diet, the presence of bacteria catalyzing TMA formation in the gut, as well as flavin monooxygenase activity in the liver [21,28]. Elevated plasma TMAO levels have been associated with alterations in reverse cholesterol transport, an increase in scavenger receptors, subsequent fat accumulation in foam cells, and the overexpression of inflammatory markers, including tumor necrosis factor-alpha (TNF-α), interleukin-6 (IL-6), and C-reactive protein (CRP), all of which promote cardiovascular disease [28].

In cirrhotic livers, FMO3 protein expression is progressively impaired with advancing disease severity [29]. In line, TMAO levels tend to be decreased in patients with end-stage liver disease compared with the general population and are inversely associated with the Model for End-stage Liver Disease (MELD) score [30]. TMAO is also decreased in patients with primary sclerosing cholangitis (PSC) and compromised liver function compared with healthy controls [31]. In such patients, relatively higher TMAO levels are associated with poorer transplantation-free survival [31]. Nevertheless, there are no data available regarding plasma TMAO concentration in LTRs. We hypothesized that plasma TMAO levels may increase after LT, partly due to the improvement in liver function.

The present study was, therefore, initiated to (i) compare plasma TMAO in stable LTRs with the levels in the general population, (ii) discern clinical-, medication-, and etiological liver disease-related factors impacting plasma TMAO, and (iii) investigate the association of plasma TMAO with all-cause mortality in LTRs.

## 2. Results

### 2.1. Baseline Characteristics and Plasma TMAO before and after Liver Transplantation

A cohort of 367 LTRs from TransplantLines and 4837 participants from the PREVEND study were included (Appendix A). The clinical and laboratory characteristics of the LTRs were obtained at a median of 8.2 (IQR 3.1–16.1) years after transplantation, as shown in Table 1. Plasma TMAO was significantly higher within the first 5 years after LT compared with subsequent years (Appendix A).

Mean age and sex distribution differed between the groups (*p* = 0.041 and *p* < 0.001, respectively), with a slightly higher age and a higher percentage of men in the LTRs (58.6%) compared with the PREVEND participants (49.4%). The distribution of BMI categories was not different between the groups. LTRs had a higher prevalence of diabetes (*p* < 0.001) and hypertension (*p* < 0.001), and a lower prevalence of current smokers compared with PREVEND participants. Alcohol consumption was substantially lower in LTRs (*p* < 0.001). eGFR was lower in LTRs (*p* < 0.001). Total cholesterol was lower but HDL cholesterol, TGs, and plasma glucose were higher in LTRs (each *p* < 0.001). The median plasma TMAO level was 4.3 (2.6–9.6) µmol/L in the LT cohort vs. 3.2 (1.7–3.6) µmol/L in the PREVEND population (*p* < 0.001), corresponding to an absolute difference of 1.1 (95% CI 0.24–2.44) µmol/L and consistent with an ~35% higher TMAO in LTRs. Notably, TMAO levels remained significantly higher in the LT group after adjusting for sex, age, BMI, hypertension, diabetes, eGFR, and statin use (*p* < 0.001). Plasma betaine was only slightly higher, resulting in a higher TMAO/betaine ratio in the LTRs (*p* < 0.001). After matching for the propensity score, 348 participants from the PREVEND cohort were included in a subsidiary analysis. It revealed that TMAO levels and the TMAO/betaine ratio remained significantly higher in the LTR group compared with the matched controls (*p* < 0.001 for each). However, differences in betaine levels were not statistically significant (*p* = 0.110).

When evaluating TMAO concentrations across different time intervals after LT, TMAO was found to be lower with increasing time after LT (Appendix A; *p* = 0.0052 using the Kruskal–Wallis test), suggesting that TMAO may decrease with increasing time after transplantation.

Of the 367 LTRs, data on plasma TMAO concentrations were available for 31 patients before and after LT (Figure 1). Among these 31 LTRs, the median TMAO concentration was 3.05 µmol/L (IQR 1.18, 4.73) pre-LT and increased to 9.25 µmol/L (IQR 3.43, 17.63) post-LT (*p* < 0.001). Notably, TMAO levels remained significantly higher in the post-LT group after adjusting for sex, age, BMI, time after transplantation, and use of glucose-lowering drugs, statins, antihypertensives, and iron supplementation (*p* = 0.017). Also, in patients in whom TMAO was measured within one year before LT (n = 30), TMAO was found to be increased after LT (10.35 µmol/L, IQR 4.48–18.33) when compared with the pre-LT measurements (3.05 µmol/L, IQR 1.10–5.08), *p* < 0.001.

### 2.2. Univariable and Multivariable Associations of TMAO with Clinical and Laboratory Variables and Medication in LTRs

As shown in Table 2, age, HbA1C, TGs, use of antihypertensives, glucose-lowering drugs, and iron supplementation, along with mycophenolate and sirolimus, were positively associated with TMAO in the LTRs, whereas HDL cholesterol, eGFR, and cyclosporin use were inversely associated with plasma TMAO.

Linear regression analysis, with adjustment for age, sex, and non-medication factors with which TMAO was associated in univariable analysis, showed that iron supplementation and sirolimus maintained positive correlations with TMAO (*p* < 0.001 and *p* = 0.044), and is presented in Appendix A.

As demonstrated in Table 3, multivariable linear regression analyses in the LTR cohort showed a positive association of TMAO with age. Fully adjusted analysis, which included age and those variables with which TMAO was associated in univariable analysis, demonstrated that TMAO was inversely associated with eGFR and positively with iron supplementation.

### 2.3. Univariable and Multivariable Associations of TMAO with Clinical and Laboratory Variables and Medication in the PREVEND Cohort

Appendix A summarizes the associations of TMAO with clinical and laboratory variables in the PREVEND cohort. TMAO was positively associated with age, fasting glucose, use of glucose-lowering drugs, and BMI, and inversely with current smoking and eGFR. In multivariable linear regression analyses, TMAO was positively associated with fasting plasma glucose and inversely with eGFR. Alcohol consumption, BMI, and lipid levels were not associated with TMAO in multivariable analysis (Appendix A).

### 2.4. Etiology of Primary Liver Disease

Table 4 provides plasma TMAO concentrations in LTRs according to the etiology of primary liver diseases.

Patients with polycystic liver disease had the highest median TMAO levels (17.80 IQR: 6.85–26.95 µmol/L) compared with other disease categories, including biliary atresia, storage diseases, autoimmune hepatitis, and cholestatic liver diseases, though not significantly different when compared with MASLD and ALD, as shown in Appendix A. Further analysis revealed that eGFR was significantly lower in patients with polycystic liver disease than in those with other etiologies (median 35.4 vs. 78.9 mL/min/1.73 m^2^, *p* < 0.001). The difference in TMAO among patients with polycystic liver disease compared with the other categories of liver disease etiologies did not remain significant after adjustment for eGFR (*p* = 0.281). Patients with MASLD and ALD as their primary etiologies also presented with significantly higher plasma TMAO concentrations compared with other etiologies (*p* < 0.001 and *p* = 0.003, respectively), while biliary atresia was associated with lower concentrations (*p* = 0.002) compared with the other etiologies.

When compared with the PREVEND cohort, TMAO concentrations were also elevated in patients with polycystic liver disease, MASLD, and ALD, as illustrated in Figure 2.

### 2.5. Longitudinal Analyses in LTRs

A total of 29 deaths (20 men and 9 women) occurred among LTRs after a median follow-up of 8.6 (IQR 3.0–16.7) years. In addition, 13 deaths were recorded in the highest TMAO tertile vs. 5 deaths in the lowest TMAO tertile. Figure 3 shows the Kaplan–Meier curves for all-cause mortality by tertiles of plasma TMAO concentrations. Mortality was highest in the highest tertile of TMAO (Log-rank test: *p* = 0.017). Appendix A shows the number of deaths according to liver disease etiology requiring transplantation stratified for TMAO tertiles.

Table 5 presents the Cox proportional hazard regression analyses with all-cause mortality, where plasma TMAO concentrations are transformed per 1 Ln SD increment.

In crude analysis, there was a positive association between TMAO per 1 Ln SD increment (HR: 1.03, 95% CI: 1.00, 1.05, *p* = 0.023), as well as with the highest TMAO tertile of TMAO compared with the lowest TMAO tertile (HR: 4.14, 95% CI: 1.47, 11.66, *p* = 0.007). This association remained significant after adjustment for sex and was close to significance after adjustment for age. This association was lost after adjustment for eGFR but was maintained after adjustment for iron supplementation and TGs, both when expressed per 1 Ln SD TMAO increment and per tertile. In crude analyses, there was an interaction of TMAO with age (*p* = 0.025) but not with sex (*p* = 0.174), impacting on all-cause mortality, However, the association of TMAO with all-cause mortality in crude analysis was confined to men (Appendix A).

Finally, among patients with MASLD as the underlying etiology of liver disease requiring transplantation, there was an association of TMAO with mortality in crude analysis both expressed per 1 LN SD increment and in the highest tertile vs. the lowest TMAO tertile (Appendix A).

## 3. Discussion

The current study is the first to comprehensively evaluate plasma TMAO concentrations in LTRs as compared with a large community-dwelling cohort, and describe the underlying factors contributing to plasma TMAO alterations. Our study reveals the following. (i) Plasma TMAO concentrations are higher in LTRs when compared with the general population, even after taking account of relevant demographic and laboratory factors, including eGFR. In line, TMAO was found to increase in a subset of patients in whom TMAO was measured before and after transplantation. (ii) There is a positive association of plasma TMAO levels with the use of iron supplementation, glucose-lowering drugs, mycophenolate, and sirolimus, and an inverse association with cyclosporin in univariable analysis of which the use of iron supplementation remained in fully adjusted multivariable analysis, which also took account of non-medication factors. (iii) Patients with polycystic liver disease, MASLD, and ALD presented with higher plasma TMAO concentrations compared with other etiologies requiring LT, as well as with the reference cohort, with the association of polycystic liver disease being lost after adjustment for the lower eGFR in this patient category. (iv) Higher plasma TMAO levels were associated with an increased risk of all-cause mortality.

Plasma TMAO concentrations were 4.3 µmol/L in the LTR cohort, with an absolute difference of 1.1 µmol/L when compared with the general population, which was significant also in a subsidiary comparison with a propensity score matching cohort. This corresponds to ~35% higher in LTRs. When compared with pre-LT measurements in a small group, TMAO increased about three-fold after LT. Such TMAO levels are higher than those reported in patients with primary sclerosing cholangitis (PSC) (median ~3 μmol/L), patients with diabetes (median 3.9 μmol/L), and patients with acute myocardial infarction (median 3.10 μmol/L) using the same NMR method [15,24,31], but are in accordance with previous reports in heart transplant recipients, measured in the chronic phase after transplantation (median 5.35 μmol/L) [32,33]. In kidney transplant recipients, the median TMAO concentration amounted to 5.66 μmol/L [23]. TMAO levels are decreased following renal transplantation [34], which is explained by the improvement of eGFR, thus enhancing renal TMAO secretion [23,34,35]. Accordingly, we found a strong inverse relationship of TMAO with eGFR in LTRs as well as in the PREVEND population, in line with our prior study [14].

Our findings in LTRs align with previous results showing a 2.4 fold increase in urinary TMAO concentrations in a small group of 40 LTRs, obtained 7 months after transplantation, compared with their levels before the procedure [36]. Complementary to the current report, we previously found that plasma TMAO concentrations tended to be lower in patients with cirrhosis compared with PREVEND participants [30]. Accordingly, circulating TMAO was lower in patients with PSC and impaired liver function compared with healthy controls and patients with ulcerative colitis [31]. The higher TMAO levels in the present study suggest that, besides expected improvement in hepatic FMO3 expression, a disrupted gut–liver axis due to chronic liver diseases and transplantation factors, such as the use of specific medications and changes in dietary patterns, could be involved in the increase of TMAO in LTRs. Plasma levels of betaine, another gut-related metabolite, were also somewhat higher in LTRs but much lower than in patients with end-stage liver disease [30]. As a result, the TMAO/betaine ratio was elevated in LTRs, opposite to our findings in patients with end-stage liver disease [30].

To explore why TMAO is elevated in LTRs, we focused on the use of medications, and found a strong, positive association with iron supplementation. There was also an association between the use of mycophenolate and sirolimus, as well as an inverse association with cyclosporin and plasma TMAO in univariate analysis. Plasma TMAO levels are affected by dietary patterns, shifts in the composition of the intestinal microflora stemming from gut dysbiosis, or disruptions of the integrity of the gut–blood barrier [20]. In a related context, balancing iron levels is crucial for preventing gut microbiota dysbiosis and inflammation. Iron supplementation influences the composition of the gut microbiota, and the intestinal solubility of oral medication plays a key role in iron absorption [37]. Given the multifactorial nature of gut health, it is plausible that alterations in the gut microbiota composition due to iron supplementation or as a result of underlying morbidities that required iron supplementation may also impact TMAO metabolism. Intriguingly, despite the absence of a relationship between TMAO and proton pump inhibitor (PPI) use, it is noteworthy that there is a relation between PPI use and iron deficiency in kidney transplant recipients [38]. Given this connection, it is plausible that iron supplementation may act as an intermediary in the complex relationship between PPI use and TMAO. PPIs have been shown to influence gut microbiota [39], and iron deficiency induced by PPI use could potentially lead to alterations in gut microbiota composition or function, impacting TMAO metabolism indirectly.

The positive univariable association observed between mycophenolate and TMAO levels in our study finds support in the documented impact of this immunosuppressant on the gastrointestinal system. Mycophenolate is known for causing gastrointestinal side effects such as diarrhea and colitis, and is important in pro-inflammatory dysbiosis and enhanced endotoxemia [40]. Mycophenolate may enhance gut permeability, potentially providing a mechanism for its positive association with TMAO. Furthermore, sirolimus may alter intestinal microbiota composition, induce abnormalities in intestinal barrier function, and cause an increase in circulating proinflammatory cytokines [41,42,43]. In humans, sirolimus is metabolized in the intestinal wall and liver, returning from the small intestine to the gut lumen via counter-transport by enterocytes, and can induce mild gastrointestinal side effects [44]. Prior reports in humans and rats have linked cyclosporin to elevated plasma and urinary TMAO levels [45,46,47]. Unexpectedly, there was an inverse univariable relationship between cyclosporin use and plasma TMAO levels. A potential explanation for this result could lie in the interplay between cyclosporin, gut microbiota, and TMAO generation. Changes in cyclosporin bioavailability, influenced by gut microbiota composition could contribute to the observed inverse association with plasma and TMAO. Of note, TMAO was only associated with sirolimus in partial regression analysis, and all associations of TMAO with the use of various immunosuppressants were lost in full multivariable analysis. Nonetheless, the mechanisms underlying the development of gut dysbiosis by the immunosuppressants used in LTRs requires further investigation that considers individual patient variability and genetic factors, conceivably contributing to diversity in TMAO responses.

Plasma TMAO concentrations are closely related to dysglycemia [48], as confirmed in the univariate analyses across both cohorts in this study. In LTRs, a positive relation was also found between TMAO and triglycerides, and a negative relation with HDL cholesterol in univariable analysis, in line with studies emphasizing the role of lipid metabolism in TMAO regulation [49,50,51].

With respect to etiologies underlying end-stage liver disease requiring transplantation, elevated TMAO levels in polycystic liver disease are likely due to the coinciding impaired kidney function, thereby compromising renal TMAO excretion [14]. MASLD has a strong association with dysglycemia and dyslipidemia [52,53]. In line with current findings, plasma TMAO was reported to be increased in PREVEND participants with an elevated fatty liver index, as a proxy of MASLD [25]. Moreover, TMAO may play role in the development and progression of MASLD by enhancing oxidative stress [54] and aggravating hepatic fat accumulation by inhibition of the farnesoid X receptor signaling [55]. In turn, hepatic steatosis may confer or be the consequence of disruptions of the integrity of the intestinal barrier [56,57]. On the other hand, alcohol is known to disrupt the gut–liver axis at multiple levels, including the gut microbiome, the intestinal epithelial barrier, and antimicrobial peptide production, which increases microbial exposure and creates a proinflammatory environment in the liver [58]. This may enhance trimethylamine (TMA) production in the gut, subsequently increasing circulating TMAO after hepatic metabolism. This conceivably explains TMAO elevations in patients with ALD [59]. The extent to which these chronic liver disease underlying etiologies could still affect circulating TMAO levels long after LT remains intriguing.

With a median time lapse of 8.2 years after transplantation as a starting point and a mean age of 55 years at the baseline evaluation, we observed 29 deaths (69% men) among 367 LTRs during 8.6 years of follow-up, corresponding to a death rate of 7.9%. This mortality rate is higher than that in the general Dutch population [60] as well as in the PREVEND cohort in which a mortality rate of 5.9% was documented after 8.3 years of follow-up [14]. Despite the relatively low number of deaths, which conforms with mortality data of other LT cohorts [61,62,63,64], TMAO was associated with all-cause mortality both when assessed by log-rank analysis and by Cox proportional hazard analysis (both per 1 Ln SD increment and per tertile to obviate bias due to potential non-linear associations). This association persisted after adjusting for sex but lost formal statistical significance following further adjustment for age, aligning with a prior report suggesting a direct age-related rise in circulating TMAO in humans due to altered microbial taxa associated with gut dysbiosis [65]. Notably, the association of TMAO with mortality was lost after adjustment for eGFR, supporting the intricate relation between TMAO and kidney function, but was maintained after adjustment for iron supplementation. A potential impact of TMAO on mortality agrees with reports in other population-based cohorts, as well as in patients with MASLD, PSC, chronic kidney disease, and chronic heart failure [14,23,25,31,66,67]. However, in patients with end-stage liver disease, we found only a borderline association of TMAO with mortality on the waiting list for LT as the outcome [30], probably consequent to a detrimental effect of TMAO per se combined with lower TMAO generation from TMA with increasing liver disease severity.

The current study has strengths and limitations. A key strength lies in its role as the first investigation to systematically evaluate the clinical impact of plasma TMAO in this patient population, including evaluation regarding all-cause mortality after LT. It also comprises robust information regarding clinical-, laboratory-, and medication-related factors in a cohort with enough of a sample size to perform detailed analyses to dissect factors associated with plasma TMAO. Several important limitations should also be acknowledged. The observational and retrospective identification of LTR participants of our study precludes making inferences regarding causality and, moreover, entails potentially confounding heterogeneity in the time period before and after LT when plasma TMAO was measured; excluding patients with a longer time after LT for TMAO measurements would decrease the sample size, follow-up time, and mortality incidence, consequently reducing statistical power. Our cross-sectional finding that TMAO may be lower with increasing time after transplantation should also be interpreted with caution; the timing of measurements could affect the interpretability of TMAO as a dynamic biomarker in the context of LT. Additionally, detailed information on diet composition was not available. Notably, in kidney transplant recipients from the same solid organ transplantation center, TMAO levels were positively associated with egg, fish, and seafood consumption, and inversely with fiber intake [23]. Moreover, the LT cohort mainly comprised individuals of north European ancestry, potentially limiting extrapolation of our findings to other ethnicities. Furthermore, individuals shortly after LT procedure were excluded, thereby obviating effects of short-term morbidities, transplant rejection issues, and intercurrent metabolic instabilities. Consequently, the clinical relevance of circulating TMAO on short-term health issues after LT remains unclear.

## 4. Materials and Methods

### 4.1. Study Design and Population

This study was carried out in accordance with STROBE (STrengthening the Reporting of OBservational studies in Epidemiology) guidelines for reporting observational studies in epidemiology [68]. The LTR cohort consisted of patients from the TransplantLines prospective observational cohort study [69], which is a comprehensive longitudinal study involving solid organ transplant recipients and donors conducted at the University Medical Center Groningen (UMCG), The Netherlands (NCT03272841). TransplantLines was approved by the Medical Ethics Committee at the UMCG, The Netherlands (METc 2014/077), and the study was performed following the guidelines of the Declaration of Helsinki [70]. All participants provided written informed consent. Patients who underwent LT up until January 2020 were included. Exclusion criteria comprised patients with no mastery of the Dutch language and with an inability to comprehend questionnaires or physical tests, those who underwent re-transplantation, and those with missing values of plasma TMAO and its related compound, betaine.

As a control population-dwelling cohort, the Prevention of REnal and Vascular ENd-stage Disease (PREVEND) study was used. This population-based cohort study was carried out among inhabitants of the city of Groningen, The Netherlands, and has been detailed elsewhere [14,71,72]. In summary, the PREVEND study was started between 1997 and 1998, and invitations to send a morning urine sample and to fill out a short questionnaire on demographics and cardiovascular history were extended to all Groningen residents aged 28 to 75 years. Pregnant women and individuals with insulin-dependent diabetes were excluded. From the first screening, all subjects with a urinary albumin concentration of ≥10 mg/L, together with a randomly selected control group with a urinary albumin concentration of <10 mg/L, were invited for further investigations in an outpatient clinic. For our analysis, we focused on participants who completed a second screening round between 2001 and 2003 that included blood and urine sample collection, were confirmed to be free of liver disease as based on a questionnaire, and had available plasma TMAO and betaine measurements. The PREVEND study was approved by the medical ethics committee at the UMCG (MEC96/01/022) and conducted following the guidelines of the declaration of Helsinki [70]. All participants from both cohorts gave written informed consent.

The primary outcome was to compare plasma TMAO concentrations between LTRs and PREVEND participants. As secondary outcomes, we (i) determined changes in TMAO before and after LT in a subset of patients, (ii) explored associated variables contributing to TMAO, including various etiologies responsible for end-stage liver disease requiring LT, and (iii) examined mortality rates in relation to TMAO concentrations.

### 4.2. Data Collection and Clinical Measurements

A schematic presentation of inclusion of LTRs and PREVEND participants is depicted in Appendix A. Patients from the TransplantLines cohort were followed from the day of transplantation until October 2020. During outpatient visits, clinical data, and blood and urine samples were collected following the TransplantLines protocol [69]. Patients maintained their regular medication during clinical evaluation and sample collection. Demographic characteristics along with data on medication use were provided by the patients and verified using electronic hospital records. Medical information including underlying disease, hospital admissions, complications after transplantation, comorbidities, and mortality was extracted from electronic hospital records. Clinical data and blood sample collections were performed between September 2016 and October 2020 involving patients that underwent LT between February 1982 and June 2020.

For the PREVEND cohort, demographic and medical information including cardiovascular disease, diabetes and renal history, and medication use was collected. Information on medication use was combined with information from a pharmacy-dispensing registry. Clinical assessment and blood sample collection from PREVEND participants were carried out between 2001 and 2003.

Anthropometric measurements, including weight, height, and body mass index (BMI in weight divided by length squared), were obtained using standardized protocols [69,71]. Alcohol consumption was quantified in grams per day, with one drink being assumed to contain 10 g of alcohol in The Netherlands, irrespective of the type of beverage [73]. Blood pressure (mmHg) was measured using an automatic device and repeated at least two times. Hypertension was defined as systolic blood pressure (SBP) > 140 mmHg, diastolic blood pressure (DBP) > 90 mmHg, or antihypertensive drug use. Diabetes was defined as fasting serum glucose > 7.0 mmol/L, non-fasting plasma glucose > 11.1 mmol/L, self-reported diagnosis, or glucose-lowering drug use. The estimated glomerular filtration rate (eGFR) was calculated using the creatinine-based 2012 CKD Epidemiology Collaboration equation [74].

### 4.3. Laboratory Measurements

The laboratory procedures for the PREVEND cohort have been previously documented in detail [14]. In both the TransplantLines and PREVEND cohorts, venous blood samples were obtained following an overnight fast. Within the TransplantLines cohort, a panel of standardized laboratory assays, including serum alanine aminotransferase (ALT), aspartate aminotransferase (AST), gamma-glutamyl transferase (GGT), alkaline phosphatase (ALP), total bilirubin, albumin, serum creatinine, high sensitivity C-reactive protein (CRP), hemoglobin, thrombocytes, leucocytes, glycated hemoglobin (HbA1C), and plasma glucose, were analyzed with standardized laboratory measurements and quality assessment control at the department of Laboratory Medicine of the UMCG.

Ethylenediaminetetraacetic acid (EDTA)-anticoagulated plasma samples were obtained by centrifugation at 1400× *g* for 15 min at 4 °C and then stored at −80 °C until analysis. PREVEND and Transplantlines samples were sent frozen at −80 °C to Labcorp, Morrisville, NC, USA. TMAO, betaine, and lipoprotein profiles in the EDTA-anticoagulated plasma samples were assessed using a Vantera^®^ Clinical Analyzer (Morrisville, NC, USA), a fully automated, high-throughput, 400 MHz proton (1H) NMR spectroscopy platform [75]. For the lipoprotein profiles, plasma samples were prepared onboard the instrument and automatically delivered to the flow probe in the NMR spectrometer’s magnetic field. Total cholesterol, high-density lipoprotein (HDL) cholesterol, and triglycerides (TGs) were quantified using the LP4 algorithm [76].

Plasma concentrations of TMAO and betaine were quantified using one-dimensional proton Carr–Purcell–Meiboom–Gill (CPMG) spectra through a deconvolution assay, as previously documented [30,72,76]. In summary, plasma samples were mixed with a citrate/phosphate buffer (3:1 v/v) to adjust the pH to 5.3, a necessary step to separate the TMAO and betaine signals that otherwise overlap at physiological pH. The betaine assay exhibited a linear range of concentrations from 26.0 to 1135 μmol/L, with coefficients of variation for intra- and inter-assay precision within 4.3% and 5.5%, respectively. The NMR-based betaine measurements showed a high level of concordance with mass spectrometry (MS) results (R^2^ =  0.94). Regarding TMAO, linearity was demonstrated up to 3000 μmol/L, with coefficients of variation of up to 4.3% for intra-assay precision and 9.8% for inter-assay precision. The NMR-based TMAO measurements exhibited strong agreement with values obtained through a mass spectrometry (MS)-based assay (R^2^ = 0.98). Prolonged storage of EDTA-anticoagulated plasma samples does not affect concentrations of TMAO, betaine, and lipid markers.

### 4.4. Statistical Analysis

Statistical analyses were conducted using the IBM SPSS software (version 28.0; IBM Corp, Armonk, NY, USA), and the Kaplan–Meier graph was generated using GraphPad Prism software (version 9.0; GraphPad Software, San Diego, CA, USA). Statistically significant results were considered if two-sided *p*-values were less than 0.05. Continuous variables are presented as mean (standard deviation) or median (interquartile range) for normally and not normally distributed data, respectively, while categorical variables are presented as numbers with percentages.

Differences in variables between LTRs and PREVEND participants were determined using T tests, Mann–Whitney U tests, and (multinomial) χ-square tests, as appropriate. To assess whether differences in TMAO levels between the groups remained significant after adjusting for covariates, the ANCOVA test was implemented. The Kruskal–Wallis test was used to calculate the mean rank of circulating TMAO in LTRs based on their primary etiology. The analysis included demographic variables (age, sex, and BMI), clinical variables (medical history and medication usage), and laboratory tests, as potential determinants. Age and sex were considered as potential confounders due to their possible associations with the primary exposure and outcomes.

Univariable relationships were assessed using Pearson correlation coefficients, calculated with 95% confidence intervals (CIs), and using TMAO and triglycerides Ln (loge) transformed to achieve approximately normal distributions. Multivariable linear regression analyses were conducted to identify independent associations of TMAO concentrations with clinical covariates, medication use, and laboratory parameters. The examination of potential interactions involved the inclusion of interaction terms in the model, statistical tests for significance, and visual inspection of interaction plots. To ensure the validity of the results, thorough model diagnostics were performed, including the assessment of linear regression assumptions through residual analysis for linearity, homoscedasticity, and independence. Normality of residuals was tested. Only variables with a variance inflation factor (VIF) lower than 5 were included, indicating acceptable levels of multicollinearity. For the propensity score matching analysis, age, sex, BMI, eGFR, lipid-lowering drugs, and antihypertensive medication were used as covariates. A match tolerance of less than or equal to 0.2 was considered to increase its accuracy.

The relationship between TMAO and time after transplantation across different time categories was determined using the Kruskal–Wallis test.

To assess the impact of the highest and lowest tertiles of TMAO on all-cause mortality after LT, Kaplan–Meier curves along with the log-rank test were used. Time-to-event Cox proportional hazards models were used to estimate hazard ratios (HRs) with 95% CIs for the association of TMAO (in tertiles and expressed per 1 Ln SD increment) with mortality after confirmation of no major departure from the proportionality of hazards assumptions [77]. HRs were calculated crude as well as with adjustments for sex, age, and other relevant variables. Interactions were sought for age and sex. Sensitivity analyses were conducted according to sex.

## 5. Conclusions

The current hypothesis-generating investigation proposes a complex regulation of TMAO in LTRs, with potential implications for metabolic health and survival. We observed elevated plasma TMAO levels in LTRs compared with a population-based cohort, which were associated with the use of specific medications and underlying liver disease etiologies that lead to end-stage liver disease requiring transplantation. In patients in whom TMAO was measured before and one year after LT, TMAO levels were also increased.

Our study was limited to a cross-sectional measurement of TMAO at various time points after transplantation, so we cannot elucidate the trajectory of TMAO changes over time and its true implications as a biomarker after LT.

The value of our study lies in serving as a starting point for future research to confirm whether TMAO is elevated after LT, identify the causes, and investigate the potential connection of this metabolite with disruptions in the gut–liver axis. This may provide valuable insights into potential therapeutic avenues to improve long-term outcomes in these patients.

## Figures and Tables

**Figure 1 ijms-25-06031-f001:**
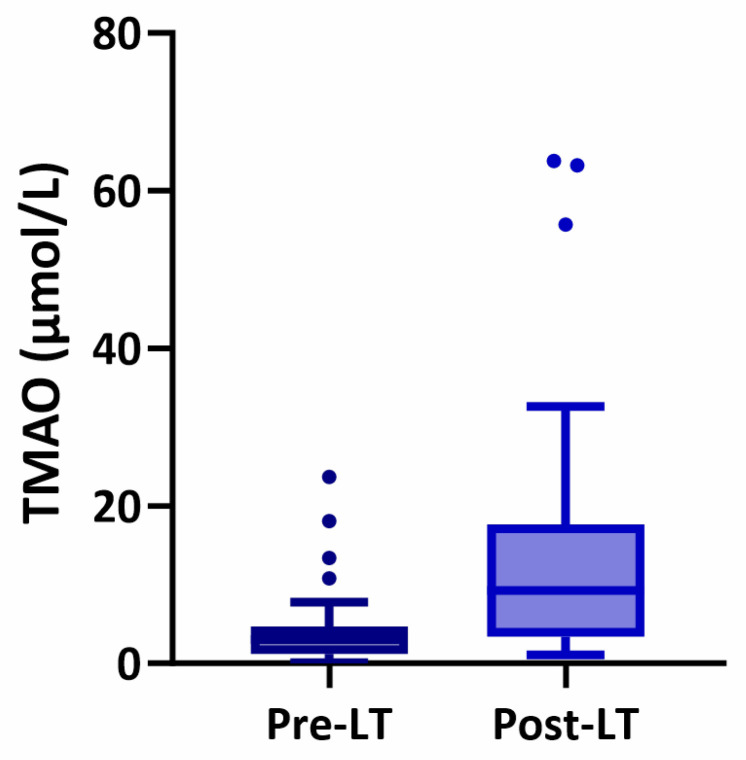
Plasma TMAO distribution among 31 patients with end-stage liver disease, before and after liver transplantation. Boxes represent the plasma TMAO concentrations (µmol/L), given as medians with interquartile ranges. Median TMAO concentration in pre-LT patients (dark blue) is 3.05 µmol/L (IQR 1.18, 4.73). Median TMAO concentration in post-LT patients (light blue) is 9.25 µmol/L (IQR 3.43, 17.63). Median time after LT: 102 days (IQR 91, 192). The dots in the graph indicate outlier data. The difference in TMAO concentration was significant (*p* < 0.001), after adjusting for sex, age, BMI, time after transplantation, and the use of glucose-lowering drugs, statins, antihypertensives, and iron supplementation (*p* = 0.017).

**Figure 2 ijms-25-06031-f002:**
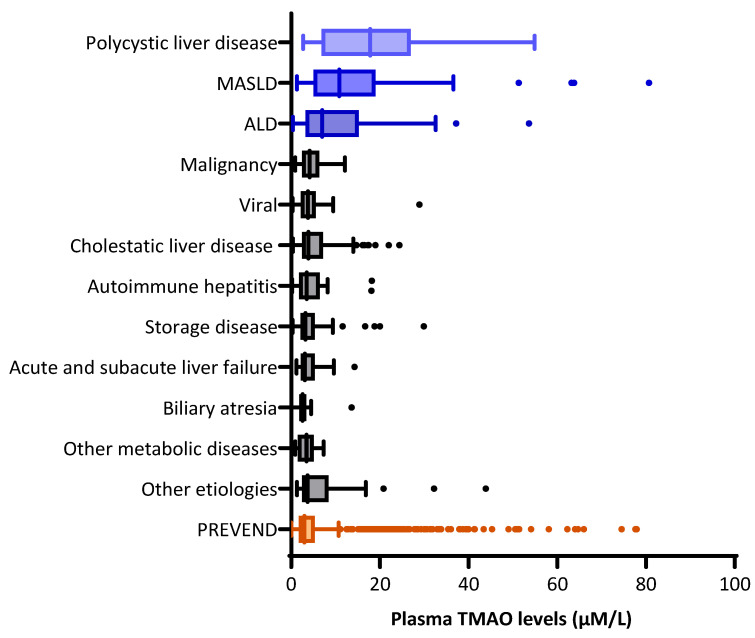
Distribution of plasma TMAO concentration in liver transplant recipients and the PREVEND population. Data are presented as medians with interquartile ranges. Etiologies with significantly high plasma TMAO concentrations are presented in blue. The PREVEND population is presented in orange. The dots in the graph indicate outlier data. Polycystic liver disease vs. PREVEND, *p* < 0.001; MASLD vs. PREVEND, *p* < 0.001; ALD vs. PREVEND, *p* < 0.001. Abbreviations: ALD, alcohol-associated liver disease; MASLD, metabolic dysfunction-associated steatotic liver disease.

**Figure 3 ijms-25-06031-f003:**
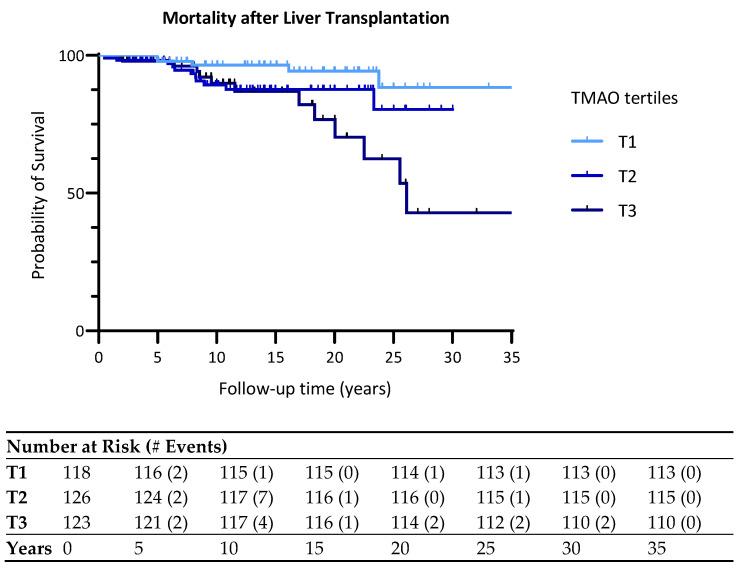
Kaplan–Meier plot for all-cause mortality following liver transplantation according to tertiles of plasma TMAO concentrations. Tertiles were defined as following: T1 < 3.2 µmol/L; T2 between 3.2 and 6.7 µmol/L; T3 > 6.7 µmol/L. Overall comparison log-rank (Mantel–Cox) test *p* = 0.017.

**Table 1 ijms-25-06031-t001:** Clinical and laboratory characteristics including TMAO in LTRs and PREVEND participants.

	Liver Transplant Recipients	PREVEND Participants	*p*-Value
	N = 367	N = 4837	
Age, years (SD)	55 (14)	54 (12)	0.041
Sex: men, n (%)	215 (58.6)	2388 (49.4)	<0.001
Height, cm (SD)	172.5 (9.9)	172.5 (9.4)	0.252
Weight, kg (SD)	79.3 (16.5)	79.7 (14.4)	0.629
BMI, kg/m^2^ (SD)	26.5 (4.9)	26.7 (4.4)	0.311
Normal: ≤25 kg/m^2^, n (%)	149 (40.6)	1882 (38.9)	0.242
Overweight: 25–30 kg/m^2^, n (%)	139 (37.9)	2043 (42.2)	
Obese: ≥30 kg/m^2^, n (%)	78 (21.3)	912 (18.9)	
Current smoking, n (%)	33 (9)	1321 (27.3)	<0.001
Alcohol consumption, g/day			<0.001
0/rarely, n (%)	258 (70.3)	1697 (35.1)	
0.1–10, n (%)	72 (19.6)	1218 (25.2)	
10–30, n (%)	25 (6.8)	993 (20.5)	
≥30, n (%)	8 (2.2)	929 (19.2)	
SBP, mmHg (SD)	133.8 (17.1)	125.8 (18.6)	0.041
DBP, mmHg (SD)	80.7 (10.8)	73.2 (8.9)	<0.001
Hypertension, n (%)	167 (45.5)	854 (17.7)	<0.001
Diabetes, n (%)	104 (28.3)	294 (6.1)	<0.001
Time after transplantation, years (IQR)	8.2 (3.1–16.1)	-	
Mortality post-transplantation, n (%)	29 (7.9)	-	
Statins, n (%)	84 (22.9)	458 (9.5)	<0.001
Antihypertensives, n (%)	167 (45.5)	854 (17.7)	<0.001
Glucose-lowering drugs, n (%)	77 (21)	178 (3.7)	<0.001
Proton pump inhibitors, n (%)	179 (48.8)	-	
Iron supplementation, n (%)	28 (7.6)	-	
Oral, n (%)	26 (7.1)	-	
Intravenous, n (%)	2 (0.5)	-	
Immunosuppressants			
Calcineurin inhibitors, n (%)	262 (71.4)	-	
Tacrolimus, n (%)	227 (61.9)	-	
Cyclosporin, n (%)	35 (9.5)	-	
Antimetabolite agents, n (%)	189 (51.5)	-	
Mycophenolate, n (%)	105 (28.6)	-	
Azathioprine, n (%)	84 (22.9)	-	
Glucocorticoids, n (%)	166 (45.2)	-	
Prednisolone, n (%)	134 (36.5)	-	
Prednisone, n (%)	32 (8.7)	-	
Sirolimus, n (%)	46 (12.5)	-	
Total cholesterol, mmol/L (IQR)	4.2 (3.6–4.9)	5.3 (4.7–6.1)	<0.001
HDL cholesterol, mmol/L (IQR)	1.3 (1.1–1.7)	1.2 (1–1.4)	<0.001
Triglycerides, mmol/L (IQR)	1.3 (1–1.8)	1.1 (0.8–1.6)	<0.001
Fasting glucose, mmol/L (IQR)	5.7 (5.2–6.9)	4.8 (4.4–5.3)	<0.001
HbA1C, mmol/mol (IQR)	36 (32–43)	-	
Hemoglobin, mmol/L (IQR)	8.5 (7.7–9.4)	8.5 (8–9)	0.738
Thrombocytes, 10^9^/L (IQR)	197 (153.3–247.5)	-	
Leucocytes, 10^9^/L (IQR)	6.1 (4.8–7.6)	-	
Albumin, g/L (IQR)	44 (42–46)	-	
CRP, mg/L (IQR)	2 (0.9–4.7)	1.3 (0.6–3)	<0.001
AST, U/L (IQR)	25 (20–33)	22 (19–26)	<0.001
ALT, U/L (IQR)	25 (18–36)	17 (13–24)	<0.001
ALP, U/L (IQR)	87 (69–127)	66 (55–79)	<0.001
Gamma-GT, U/L (IQR)	40.5 (21–89.5)	24 (16–38)	<0.001
Total bilirubin, µmol/L (IQR)	10 (7–14)	7 (5–9)	<0.001
Serum creatinine, µmol/L (IQR)	88.9 (74.5–107.5)	83.2 (73.9–92.4)	<0.001
eGFR, mL/min (IQR)	77.9 (60.9–97.8)	93.7 (81.6–104.3)	<0.001
TMAO, µmol/L (IQR)	4.3 (2.6–9.6)	3.2 (1.7–5.6)	<0.001
Betaine, µmol/L (IQR)	38.8 (33.4–46.8)	36.7 (30.7–43.6)	<0.001
TMAO/Betaine ratio (IQR)	0.11 (0.06–0.25)	0.09 (0.04–0.16)	<0.001

Data are given in numbers with percentages (%), mean with standard deviation (SD) or median with interquartile ranges (IQR). Abbreviations: BMI, body-mass index; SBP, systolic blood pressure; DBP, diastolic blood pressure; LDL, low-density lipoprotein; HDL, high-density lipoprotein; HbA1C, glycated hemoglobin; CRP, C-reactive protein; AST, aspartate aminotransferase; ALT, alanine aminotransferase; ALP, alkaline phosphatase; Gamma-GT, gamma-glutamyl transferase; eGFR, estimated glomerular filtration rate; TMAO, trimethylamine N-oxide.

**Table 2 ijms-25-06031-t002:** Pearson correlation coefficients between clinical and laboratory variables and TMAO in LTRs.

	TMAO (µmol/L)	
	Pearson Correlation Coefficient	*p*-Value
Age	0.301	<0.001
Sex	0.002	0.969
BMI	0.089	0.089
Current smoking	−0.034	0.565
Alcohol consumption	−0.008	0.890
SBP	0.103	0.064
HbA1C	0.146	0.006
Fasting glucose	0.126	0.020
Total cholesterol	0.089	0.087
HDL cholesterol	−0.111	0.034
Triglycerides	0.247	<0.001
eGFR	−0.428	<0.001
Antihypertensives	0.122	0.021
Glucose-lowering drugs	0.193	<0.001
Statins	0.011	0.838
PPIs	0.093	0.079
Iron supplementation	0.169	<0.001
Calcineurin inhibitors	−0.044	0.411
Tacrolimus	0.028	0.596
Cyclosporin	−0.116	0.028
Antimetabolite agents	0.029	0.579
Mycophenolate	0.127	0.016
Azathioprine	−0.097	0.066
Glucocorticoids	−0.033	0.533
Prednisolone	−0.052	0.325
Prednisone	0.031	0.562
Sirolimus	0.135	0.011

TMAO was Ln transformed for correlation analysis. Abbreviations: BMI, body-mass index; SBP, systolic blood pressure; HbA1C, glycated hemoglobin; HDL, high-density lipoprotein; eGFR, estimated glomerular filtration rate; PPIs: proton-pump inhibitors.

**Table 3 ijms-25-06031-t003:** Multivariable linear regression analyses demonstrating independent associations of circulating TMAO in liver transplant recipients.

	Model 1		Model 2		Model 3	
	Std. β (95% CI)	*p*-Value	Std. β (95% CI)	*p*-Value	Std. β (95% CI)	*p*-Value
Age	0.214 (0.112, 0.316)	<0.001	−0.080 (−0.197, 0.037)	0.184	−0.045 (−0.164, 0.074)	0.460
HbA1C			0.037 (−0.065, 0.139)	0.481	0.008 (−0.121, 0.136)	0.903
HDL cholesterol			−0.057 (−0.155, 0.041)	0.256	−0.075 (−0.174, 0.025)	0.141
Triglycerides			0.118 (0.014, 0.222)	0.027	0.097 (−0.009, 0.203)	0.075
eGFR			−0.460 (−0.572, −0.347)	<0.001	−0.429 (−0.545, −0.313)	<0.001
Antihypertensives					−0.046 (−0.146, 0.054)	0.369
Glucose-lowering drugs					0.052 (−0.071, 0.175)	0.409
Cyclosporin					−0.058 (−0.151, 0.035)	0.223
Mycophenolate					−0.020 (−0.118, 0.078)	0.688
Sirolimus					0.035 (−0.059, 0.129)	0.469
Iron supplementation					0.130 (0.034, 0.226)	0.008

Std. β: standardized regression coefficients. Model 1: adjusted for age. Model 2: Model 1 + HbA1C, HDL cholesterol, triglycerides, and eGFR. Model 3: Model 2 + antihypertensives, glucose-lowering drug, cyclosporin, mycophenolate, sirolimus and iron supplementation. Abbreviations: HbA1C, glycated hemoglobin; HDL, high-density lipoprotein; eGFR, estimated glomerular filtration rate.

**Table 4 ijms-25-06031-t004:** Distribution of plasma TMAO concentrations according to the etiology of primary liver diseases in the liver transplant recipients.

Etiology of Primary Liver Disease	*N*	TMAO (µmol/L)	*p*-Value
Cholestatic liver disease	100	4.00 (2.40–7.55)	0.193
ALD	48	7.00 (3.23–15.23)	0.003
Viral	37	3.80 (2.20–5.55)	0.069
MASLD	34	10.85 (5.00–18.98)	<0.001
Storage disease	33	3.30 (2.15–8.50)	0.288
Autoimmune hepatitis	19	3.60 (1.80–7.30)	0.231
Acute and subacute liver failure	18	3.10 (2.18–6.88)	0.158
Biliary atresia	16	2.55 (1.83–3.43)	0.002
Polycystic liver disease	13	17.80 (6.85–26.95)	<0.001
Malignancy	10	4.15 (2.40–6.35)	0.496
Other metabolic diseases	6	3.45 (1.50–5.13)	0.209
Other	33	4.10 (2.63–8.85)	0.701

Data are presented as medians with interquartile ranges. *p*-values using the Mann–Whitney U test, compared with other etiologies of primary liver diseases combined, Abbreviations: ALD, alcohol-associated liver disease; MASLD, metabolic dysfunction associated steatotic liver disease.

**Table 5 ijms-25-06031-t005:** Association of TMAO per 1 Ln SD increment with all-cause mortality in liver transplant recipients.

	TMAO per 1 Ln SD Increment		T1	T2		T3	
Participants, n	367		118	126		123	
Events, n	29		5	11		13	
	HR (95% CI)	*p*-Value		HR (95% CI)	*p*-Value	HR (95% CI)	*p*-Value
Crude model	1.03 (1.00, 1.05)	0.023	(ref)	2.50 (0.87, 7.21)	0.090	4.14 (1.47, 11.66)	0.007
Model 1	1.03 (1.00, 1.05)	0.025	(ref)	2.46 (0.85, 7.09)	0.097	3.96 (1.40, 11.18)	0.009
Model 2	1.02 (0.99,1.04)	0.139	(ref)	1.96 (0.66, 5.82)	0.226	2.79 (0.94, 8.33)	0.065
Model 3	1.00 (0.96, 1.03)	0.900	(ref)	1.54 (0.56, 4.22)	0.403	1.66 (0.57, 4.87)	0.357
Model 4	1.03 (1.00, 1.05)	0.024	(ref)	2.15 (0.79, 5.82)	0.133	3.39 (1.28, 8.95)	0.014
Model 5	1.02 (1.00, 1.05)	0.050	(ref)	2.36 (0.82, 6.83)	0.113	3.52 (1.04, 11.87)	0.043
Model 6	1.00 (0.98, 1.03)	0.743	(ref)	1.71 (0.57, 5.07)	0.333	1.60 (0.44, 5.87)	0.480

Data are presented as hazard ratios with 95% confidence intervals and *p* values. Model 1: crude model + sex. Model 2: crude model + age. Model 3: crude model + eGFR. Model 4: crude model + iron supplementation. Model 5: crude model + triglycerides. Model 6: crude model + sex + eGFR + iron supplementation + triglycerides. Abbreviation: eGFR, estimated glomerular filtration rate.

## Data Availability

The anonymized raw data supporting the conclusions of this article will be made available by the authors on request.

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
