# Peer review of "Circulating Trimethylamine-N-Oxide Is Elevated in Liver Transplant Recipients"

_ijms, 2024, doi:10.3390/ijms25116031_

Round 1
Reviewer 1 Report
Comments and Suggestions for Authors
This study collected and analysed plasma TMAO in LTR and some participants selected from the PREVEND study. It reports that TMAO is elevated in stable LTR compared with PREVEND participants. In LTR, TMAO levels were negatively associated with eGFR and positively associated with iron supplementation. Also, plasma TMAO may predict the mortality risk in LTR. Here are some comments:
1. PREVEND participants were just used as "normal TMAO" controls. These data should be used more wisely, e.g., to smartly explain what you found in the LTR - LTR due to ALD have a higher TAMO (Table 4), but Alcohol consumption was not correlated with TAMO in LTR (Table 2), how about it in the PREVEND participants? indicating the design of this study should be improved, e.g., "nest-control", "paired based on certain conditions", etc. in order to find out what are LTR-specific factors affecting TAMO
2. The Fig. 2 data are interesting: have you considered the distribution of sex and age in T1, T2 and T3? How about the result after adjusting sex and age? Can eGFR, Iron intake, TG, etiologies, or others predict the mortality and difference compared to TAMO?
3. From Fig. 2, TAMO was higher in the first 5 years post-LT than others (to be confirmed). Can you look at the data (typically the TAMO. And other analysis if you want) based on the follow-up time?
4. The discussion is very long, though supplying many TAMO-related information, but a large part of them are not directly related with the data shown in the current study.
5. Conclusions (lines 496-507): the majority of conclusions are not supported by your data. For example,
"We demonstrated plasma TMAO elevations after LT..." is not supported by the data. "plasma TMAO elevations" in these LTRs, but could happen before LT.The conclusion should be re-written.
6. minor: Table 1, what is the number inside the ()? SD? SEM? or interquartile (then %?)?
Reviewer 2 Report
Comments and Suggestions for Authors
The article “Circulating Trimethylamine-N-Oxide and Mortality in Liver 2 Transplant Recipients” has opted to assess the possible relation between biomarkers and long-term survival rates of liver transplant recipients (LTR). The basis of checking the biomarkers rests on the potential relation of survival of LTR patients with metabolic and cardiovascular pathologies. In this regard, the authors checked a potential biomarker related to the gut microbiome-derived metabolite, trimethylamine 51 N-oxide (TMAO, N, N-dimethylmethanamine N-oxide). TMAO was measured in the sera of two cohorts: A stable LTR from the Transplant-Lines cohort and in participants from the population-based PREVENT cohort. TMAO was elevated in LTR with metabolic dysfunction-associated steatotic liver disease, alcoholic liver disease, and polycystic disease as underlying etiology. Among LTR, TMAO levels were independently associated with eGFR and iron supplementation and were associated with mortality. This led to the conclusion that plasma TMAO is elevated in stable LTR, with impaired eGFR and iron supplementation as potential contributory factors. The authors also conclude that TMAO may be related to increased mortality in such patients.
Comments
1. The role of TMAO in liver diseases is a known factor. Elevated TMAO has been reported in liver cirrhosis, and the relation of TMAO with the severity of liver cirrhosis has been shown (van den Berg EH et al. High plasma levels of betaine, a trimethylamine N-Oxide-related metabolite, are associated with the severity of cirrhosis. Liver Int. 2023;43:424-433)
2. Also, the role of TMAO has been shown in the prognosis of chronic liver disease (Kummen et al. Elevated trimethylamine-N-oxide (TMAO) is associated with poor prognosis in primary sclerosing cholangitis patients with normal liver function. Hov JR. United European Gastroenterol J. 2017 Jun;5(4):532-541).
3. Taking the two above descriptive comments and observations, the article's novelty is limited as the present study has attempted to assess the role of TMAO in another liver pathology.
4. From this study, it remains elusive if increased TMAO is related to poor prognosis of transplanted subjects or if these have any prognostic importance.
5. The study is retrospective, and a prospective study, even with a small sample size, is warranted to properly exhibit the role of TMAO, if any.
6. I am unsure when sera were collected for measuring TMAO.
7. The kinetics of TMAO at different points must be studied in this type of article. Instead, I would like to know if TMAO is elevated from the beginning of these patients or if the time of taking samples played a role in this context.
Round 2
Reviewer 2 Report
Comments and Suggestions for Authors
The article in question noted a biomarker that may be relevant during the clinical assessment of the prognosis of liver transplantation. The prognosis of liver transplantation is a combination of several complex mechanisms. The authors have worked with TMAO, and several studies have already been accomplished. After an initial review of the article, I provided several comments, and the authors replied to these queries from their end.
Comments
1. I have emphasized the time of estimation of TMAO in their cohort. They responded that TMAO was measured in September 2016 and October 2020 in patients who underwent LT between February 1982 and June 2020. Thus, there is a lack of proper data timing. In other words, TMAO was not measured just before liver transplantation in each patient. If that is the case, the measurement of TMA at some point has almost no clinical implication. The authors tried to resolve this query by providing data from 31 patients in whom TMAO data before, that is, with end-stage liver disease and after LT (see below). (Lines 124 - 129, 248 - 250, 262 – 263, 537 - 538, and new Figure 1).
2. 2. In addition, the data of TMA at different points in standard control subjects and patients with LT and patients not undergoing LT should be measured to validate if TMAO represents any serum marker. The authors take the lesson by analyzing the development of SGPT measurement for liver damage from a historical perspective.
3. Finally, I assume that if TMAO is considered as a biomarker of prognosis of LT, the study design should be accomplished appropriately with different controls in a prospective study.
Round 3
Reviewer 2 Report
Comments and Suggestions for Authors
The authors have responded twice to emphasize the role of TMAO; however, the fundamental issues still need to be resolved. I have mentioned these, and the responses are not satisfactory, so the conclusion shown in the last sentence of the abstract: “Plasma TMAO may contribute to increased mortality risk in such patients,” can be validated. Proposing a biomarker for the prognosis of LT requires significant validation and proper study design.
